# Short communication: Inverse isochron regression for Re–Os, K–Ca and other chronometers

Yang Li[1] and Pieter Vermeesch[2]

[1]State Key Laboratory of Lithospheric Evolution, Institute of Geology and Geophysics, Chinese Academy of Sciences, Beijing 100029

[2]Department of Earth Sciences, University College London, Gower Street, London WC1E 6BT

**Correspondence:** Pieter Vermeesch (p.vermeesch@ucl.ac.uk)

**Abstract.** Conventional Re–Os isochrons are based on mass spectrometric estimates of $^{187}$Re/$^{188}$Os and $^{187}$Os/$^{188}$Os, which often exhibit strong error correlations that may obscure potentially important geological complexity. Using an approach that is widely accepted in $^{40}$Ar/$^{39}$Ar and U–Pb geochronology, we here show that these error correlations are greatly reduced by applying a simple change of variables, using $^{187}$Os as a common denominator. Plotting $^{188}$Os/$^{187}$Os vs. $^{187}$Re/$^{187}$Os produces an 'inverse isochron', defining a binary mixing line between an inherited Os-component whose $^{188}$Os/$^{187}$Os-ratio is given by the vertical intercept, and the radiogenic $^{187}$Re/$^{187}$Os-ratio, which corresponds to the horizontal intercept. Inverse isochrons facilitate the identification of outliers and other sources of data dispersion. They can also be applied to other geochronometers such as the K–Ca method and (with less dramatic results) the Rb–Sr, Sm–Nd and Lu–Hf methods. Conventional and inverse isochron ages are similar for precise datasets, but may significantly diverge for imprecise ones. A semi-synthetic data simulation indicates that, in the latter case, the inverse isochron age is more accurate. The generalised inverse isochron method has been added to the `IsoplotR` toolbox for geochronology, which automatically converts conventional isochron ratios into inverse ratios and vice versa.

## 1  Introduction: the conventional Re–Os isochron

The [$^{187}$Os/$^{188}$Os]-budget of a $^{187}$Re-bearing rock or mineral can be divided into an inherited component and a radiogenic component:

$$\left[\frac{^{187}\text{Os}}{^{188}\text{Os}}\right] = \left[\frac{^{187}\text{Os}}{^{188}\text{Os}}\right]_i + \left[\frac{^{187}\text{Re}}{^{188}\text{Os}}\right](\exp[\lambda_{187}t]-1) \tag{1}$$

where $\lambda_{187}$ is the decay constant of $^{187}$Re ($= 1.666 \times 0.017$ yr$^{-11}$, Smoliar et al., 1996) and $t$ is the time elapsed since isotopic closure. Equation 1 forms the equation of a line:

$$y = a + bx \tag{2}$$

where $x = \left[^{187}\text{Re}/^{188}\text{Os}\right]$ $y = \left[^{187}\text{Os}/^{188}\text{Os}\right]$, $a = \left[^{187}\text{Os}/^{188}\text{Os}\right]_i$ and $b = (\exp[\lambda_{187}t]-1)$. Both the independent variable ($x$) and the dependent variable ($y$) are measured quantities that are associated with analytical uncertainty. Therefore, linear

regression of the isochron line is typically done by weighted least squares regression with uncertainty in both variables (York et al., 2004).

One drawback of the conventional isochron definition of Equation 1 is that the rarest isotope, $^{188}$Os, which is associated with the largest mass spectrometer uncertainties, appears in the denominator of both $x$ and $y$. This has the potential to produce strong error correlations (Stein et al., 2000). For example, consider the following hypothetical (independent) abundance estimates and their standard errors:

$$X \equiv {}^{187}\text{Os} = 2,000 \pm 10 \text{ fmol}; \ Y \equiv {}^{187}\text{Re} = 30,000 \pm 50 \text{ fmol and } Z \equiv {}^{188}\text{Os} = 10 \pm 2 \text{ fmol}$$

then, using the methods of Pearson (1896), the ratio correlation between [$^{187}$Os/$^{188}$Os] and [$^{187}$Re/$^{188}$Os] is

$$\rho_{\frac{X}{Z}\frac{Y}{Z}} \approx \frac{\left(\frac{s[Z]}{Z}\right)^2}{\sqrt{\left(\frac{s[Y]}{Y}\right)^2 + \left(\frac{s[Z]}{Z}\right)^2}\sqrt{\left(\frac{s[X]}{X}\right)^2 + \left(\frac{s[Z]}{Z}\right)^2}} = \frac{\left(\frac{2}{10}\right)^2}{\sqrt{\left(\frac{50}{30,000}\right)^2 + \left(\frac{2}{10}\right)^2}\sqrt{\left(\frac{10}{2,000}\right)^2 + \left(\frac{2}{10}\right)^2}} = 0.9997 \qquad (3)$$

The strong error correlation between the two variables on the isochron diagram is manifested as narrow and steeply inclined error ellipses, which may graphically obscure any geologically significant trend.

As an example, consider the Re–Os dataset of Morelli et al. (2007) (Figure 1a), which represents a mixture of three samples. At first glance, this dataset appears to define an excellent isochron with a clear slope corresponding to an isochron age of 287 Ma. However upon closer inspection, the interpretation of this fit is not so simple:

1. The error ellipses exhibit a tremendous range of sizes. The plot is dominated by the least precise measurement (i.e. aliquot 14), and the remaining aliquots are barely visible.

2. The error ellipses are nearly perfectly aligned with the isochron, which makes it difficult to distinguish between geological and analytical sources of correlation.

3. The isochon fit exhibits an MSWD of 2.5, which indicates the presence of a moderate amount of overdispersion of the data with respect to the formal analytical uncertainties. It is not immediately clear which aliquots are responsible for the poor goodness-of-fit.

## 2 The inverse Re–Os isochron

All three of these problems can be solved by a simple change of variables:

$$\left[\frac{^{188}\text{Os}}{^{187}\text{Os}}\right] = \left[\frac{^{188}\text{Os}}{^{187}\text{Os}}\right]_i \left\{1 - \left[\frac{^{187}\text{Re}}{^{187}\text{Os}}\right](\exp[\lambda_{187}t] - 1)\right\} \qquad (4)$$

which defines an 'inverse' isochron line:

$$y' = a' + b'x' \qquad (5)$$

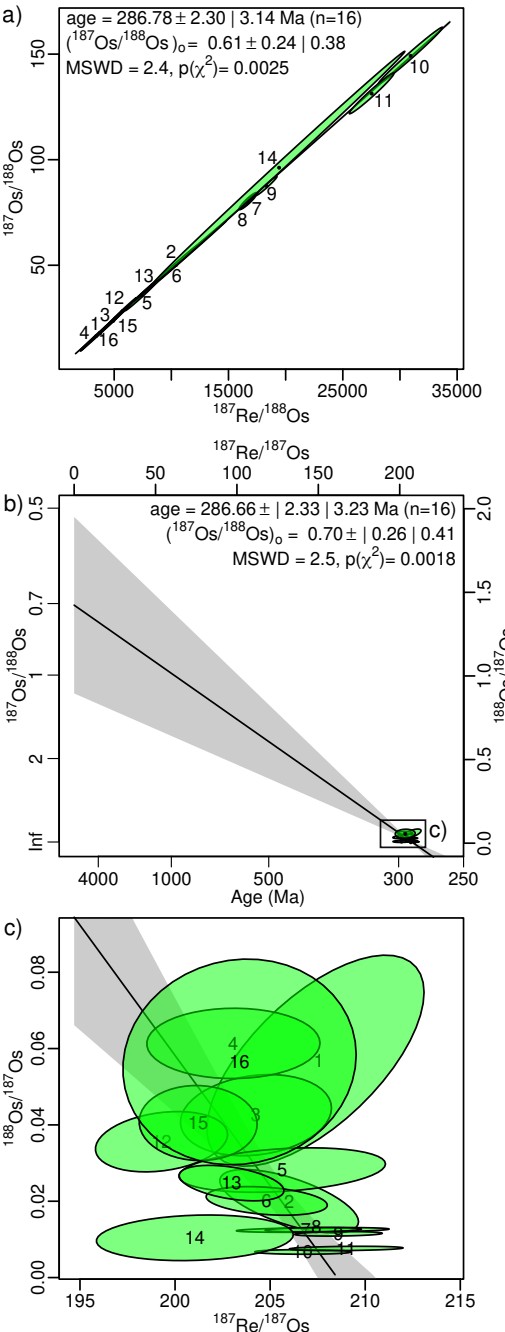

**Figure 1.** a) conventional isochron of the Re–Os data for Morelli et al. (2007), with uncertainties shown as 95% confidence intervals without and with $\sqrt{\mathrm{MSWD}}$ overdispersion multiplier. b) and c) the inverse isochron diagram of the same data represents a mixing line between inherited and radiogenic components. The sample is highly radiogenic, allowing precise age estimation despite the presence of significant overdispersion, which is masked by the error correlations in the conventional isochron diagram. All error ellipses and confidence envelopes are shown at 95% confidence.

where $x' = \left[^{187}\mathrm{Re}/^{187}\mathrm{Os}\right]$, $y' = \left[^{188}\mathrm{Os}/^{187}\mathrm{Os}\right]$, $a' = \left[^{188}\mathrm{Os}/^{187}\mathrm{Os}\right]_i$ and $b' = -\left[^{188}\mathrm{Os}/^{187}\mathrm{Os}\right]_i \left(\exp[\lambda_{187}t] - 1\right)$.

Equation 4 defines a mixing line between the non-radiogenic $\left[^{188}\mathrm{Os}/^{187}\mathrm{Os}\right]$-ratio (which marks the vertical intercept) and the radiogenic $\left[^{187}\mathrm{Re}/^{187}\mathrm{Os}\right]$-ratio (which markes the horizontal intercept). By moving the least abundant nuclide to the numerator of the dependent variable, instead of the denominator of both the dependent and the independent variables, the inverse isochron reduces the error correlations. Revisiting the earlier hypothetical example yields an error correlation of:

$$\rho_{\frac{Y}{X}\frac{Z}{X}} \approx \frac{\left(\frac{s[X]}{X}\right)^2}{\sqrt{\left(\frac{s[X]}{X}\right)^2 + \left(\frac{s[Y]}{Y}\right)^2}\sqrt{\left(\frac{s[X]}{X}\right)^2 + \left(\frac{s[Y]}{Y}\right)^2}} = \frac{\left(\frac{10}{2,000}\right)^2}{\sqrt{\left(\frac{10}{2,000}\right)^2 + \left(\frac{50}{30,000}\right)^2}\sqrt{\left(\frac{10}{2,000}\right)^2 + \left(\frac{2}{10}\right)^2}} = 0.024 \qquad (6)$$

Plotting the Morelli et al. (2007) dataset on an inverse isochron diagram provides a much clearer picture of it (Figure 1b and c):

1. Although the error ellipses still exhibit a range of sizes, reflecting the heteroscedasticity of the data, the imprecise measurements do no longer dominate the plot to the extent where they obscure the precise ones.

2. The error ellipses are no longer aligned parallel to the isochron line, but are oriented at an angle to it. This makes it easier to see the difference between the geological and analytical sources of correlation.

3. The overdispersion is clearly visible and can be attributed to aliquots 1, 12 and 14, whose error ellipses exhibit the smallest overlap with the best fit line. Most of the geochronologically valuable information is contained in the highly radiogenic aliquots 7–11, which tightly cluster near the $[^{187}\mathrm{Re}/^{187}\mathrm{Os}]$-intercept. Even though the data are overdispersed, the overall composition is very radiogenic and can therefore be used to obtain precise age constraints. The initial $[^{187}\mathrm{Os}/^{188}\mathrm{Os}]$-ratio, however, is poorly constrained.

# 3   Application to other chronometers

Strong error correlations are commonly observed in other conventional isochron systems, where they may arise from a number of mechanisms including poor counting statistics (previous sections), blank correction (e.g., Vermeesch, 2015; Connelly et al., 2017) or fractionation (e.g., Ludwig, 1980). Inverse isochron ratios, in which the radiogenic daughter isotope is used as a common denominator, are commonplace in $^{40}\mathrm{Ar}/^{39}\mathrm{Ar}$ (Turner, 1971) and U–Pb (Tera and Wasserburg, 1972) geochronology. They are equally applicable to other dating methods, such as Rb–Sr ($[^{87}\mathrm{Sr}/^{86}\mathrm{Sr}]$ vs. $[^{87}\mathrm{Rb}/^{86}\mathrm{Sr}]$), Sm–Nd ($[^{144}\mathrm{Nd}/^{143}\mathrm{Nd}]$ vs. $[^{147}\mathrm{Sm}/^{143}\mathrm{Nd}]$), Lu–Hf ($[^{177}\mathrm{Hf}/^{176}\mathrm{Hf}]$ vs. $[^{176}\mathrm{Lu}/^{176}\mathrm{Hf}]$) and K–Ca ($[^{44}\mathrm{Ca}/^{40}\mathrm{Ca}]$ vs. $[^{40}\mathrm{K}/^{40}\mathrm{Ca}]$).

In the case of K–Ca dating, the inverse approach offers similar benefits as for the Re–Os method because $^{44}\mathrm{Ca}$ is typically 100 times less abundant than $^{40}\mathrm{Ca}$, thus making the conventional isochron plot prone to strong error correlations. Note that some K–Ca studies use $^{42}\mathrm{Ca}$ as a normalising isotope, which is even less abundant $^{44}\mathrm{Ca}$, and therefore further aggravates the problem. For other chronometers such as Rb–Sr, Sm–Nd and Lu–Hf, whose non-radiogenic isotopes are at least as abundant as the radiogenic daughter isotopes, the benefits of the inverse isochron approach are less obvious.

Given a data table of conventional isochron ratios ($x$ and $y$ in Equation 2), it is possible to calculate the inverse ratios ($x'$ and $y'$ in Equation 5), their uncertainties ($s[x']$ and $s[y']$) and error correlations ($\rho_{x'y'}$) using the following equations:

$$
\begin{cases}
x' = \frac{x}{y} \\
y' = \frac{1}{y} \\
\left(\frac{s[x']}{x'}\right)^2 = \left(\frac{s[x]}{x}\right)^2 - 2\rho_{x,y}\left(\frac{s[x]}{x}\right)\left(\frac{s[y]}{y}\right) + \left(\frac{s[y]}{y}\right)^2 \\
\left(\frac{s[y']}{y'}\right)^2 = \left(\frac{s[y]}{y}\right)^2 \\
\rho_{x'y'} = \left(\frac{x'}{s[x']}\right)\left[\left(\frac{y}{s[y]}\right) - \rho_{xy}\left(\frac{x}{s[x]}\right)\right]
\end{cases}
\tag{7}
$$

This transformation is perfectly symmetric in the sense that it can also be used to convert inverse isochron ratios to conventional ones. To do this, it suffices to swap $x'$ and $y'$ for $x$ and $y$ and vice versa.

## 4 A semi-synthetic test of accuracy

Dalrymple et al. (1988) assert that conventional and inverse isochron regression are mathematically equivalent in the context of $^{40}$Ar/$^{39}$Ar geochronology. This is indeed the case when the analytical uncertainties of the parent-daughter ratios are relatively small ($< 5\%$, say), as is the case for the Re–Os example of Figure 1. However, this is no longer true when the analytical uncertainties are large, or when the data are significantly dispersed around the best fitting isochron line. In those cases the conventional and inverse isochrons can yield substantially different age estimates. This is because isotopic ratios are strictly positive quantities with skewed error distributions, and the weighted least squares algorithm of York et al. (2004) does not take into account this skewness.

A full theoretical discussion of this phenomenon falls outside the scope of our short communication. Instead, we will compare and contrast the accuracy of conventional and inverse isochrons using a semi-synthetic dataset based on 30 K–Ca ion microprobe measurements published by Harrison et al. (2010):

1. Let $x_i$ be the $i^{\text{th}}$ $^{40}$K/$^{44}$Ca ratio measurement, and let $\sigma[x_i]$, $\sigma[y_i]$, $\rho[x_i, y_i]$ be the standard errors and error correlation of the corresponding $^{40}$K/$^{44}$Ca and $^{40}$Ca/$^{44}$Ca ratios.

2. Collect $n$ pairs of logratios $\{\ln[X_i], \ln[Y_i]\}$ from a bivariate normal distribution with means $\{\ln[x_i], \ln[y_i]\}$ and covariance matrix $\Sigma_i$ where

$$
y_i = y_\circ + 0.895 x_i (\exp[\lambda_{40} t] - 1)
\tag{8}
$$

in which $y_\circ = 66$ is the initial $^{40}$Ca/$^{44}$Ca ratio, $t = 800$ Ma is the true K–Ca age, and

$$
\Sigma_i = \begin{bmatrix} \frac{1}{x_i} & 0 \\ 0 & \frac{1}{y_i} \end{bmatrix} \begin{bmatrix} \sigma[x_i]^2 & \rho[x_i, y_i]\sigma[x_i]\sigma[y_i] \\ \rho[x_i, y_i]\sigma[x_i]\sigma[y_i] & \sigma[x_i]^2 \end{bmatrix} \begin{bmatrix} \frac{1}{x_i} & 0 \\ 0 & \frac{1}{y_i} \end{bmatrix}
$$

3. The semi-synthetic dataset is then given by $\{X_i, Y_i\}$ (for $1 \leq i \leq n$) with covariance matrices $\Sigma_i'$ that are computed as follows:

$$\Sigma_i' = \begin{bmatrix} X_i & 0 \\ 0 & Y_i \end{bmatrix} \Sigma_i \begin{bmatrix} X_i & 0 \\ 0 & Y_i \end{bmatrix}$$

The logarithmic transformation is necessary to account for the inevitable skewness of the error distributions. Even though the semi-synthetic dataset is defined in terms of the conventional isochron equation (Eq. 8), Figure 2 shows that it is the inverse isochron that most accurately estimates the age. We therefore recommend that inverse isochrons replace conventional isochrons in Re–Os and K–Ca geochronology. The difference between the conventional and inverse isochron age may serve as a measure of robustness for the results.

## 5  Implementation in `IsoplotR`

Inverse isochrons have been added to all the relevant chronometers in the `IsoplotR` toolbox for radiometric geochronology (Vermeesch, 2018). This functionality can be used either from the graphical user interface (which can be accessed both online and offline, Figure 3a), or from the command line, using the `R` programming language and application programming interface (Figure 3b and c). `IsoplotR` automatically executes the ratio conversion of Equation 7 in the background, so the user can supply their data as conventional ratios and still plot them on an inverse isochron diagram.

## 6  Conclusions

Conventional isochrons are straight line regressions between two ratios $D/d$ and $P/d$, where $P$ and $D$ are the parent and daughter nuclides, and $d$ is a non-radiogenic isotope of the daughter element. This paper reviewed the phenomenon whereby strong error correlations arise when $d$ is less abundant than $D$, and is therefore measured less precisely than $D$. This is the case in Re–Os and K–Ca geochronology, which use $^{188}$Os and $^{44}$Ca as normalising isotopes, respectively. These isotopes are tens to hundreds of times less abundant than the radiogenic $^{187}$Os and $^{40}$Ca, causing strong error correlations. Besides this 'spurious' source of correlated uncertainties (*sensu* Pearson, 1896), additional sources of covariance may include blank corrections, calibrations and fractionation effects that apply to both variables in the isochron regression.

The error correlation between the isochron ratio measurements can be so strong ($r > 0.99$) that it outweighs and obscures the geochronological correlation. This is not only inconvenient from an esthetic point of view, but may also cause numerical problems. It is not uncommon for data tables to either not report error correlations at all, or to report them to only one significant digit. However, the difference between error correlations of $r = 0.991$ and $r = 0.999$, say, may have a large effect on the isochron age. All these problems can be solved by recasting the isochron regression into a new form, by plotting $d/D$ vs. $P/D$. This produces a different type of linear trend, in which the vertical intercept yields the reciprocal daughter ratio, and the age is not proportional to the slope of the isochron line, but inversely proportional to its horizontal intercept.

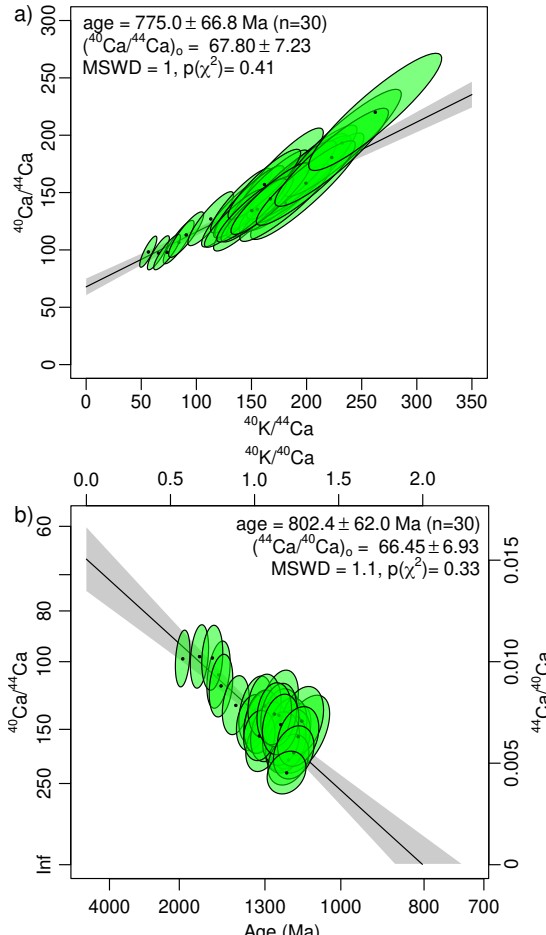

**Figure 2.** a) conventional and b) inverse isochron for a representative outcome of the semi-synthetic K–Ca data generator. The true age is 800 Ma and the true initial $^{40}Ca/^{44}Ca$ ratio is 66. The inverse isochron better approximates these values than the conventional isochron. Uncertainties are shown as 95% confidence intervals.

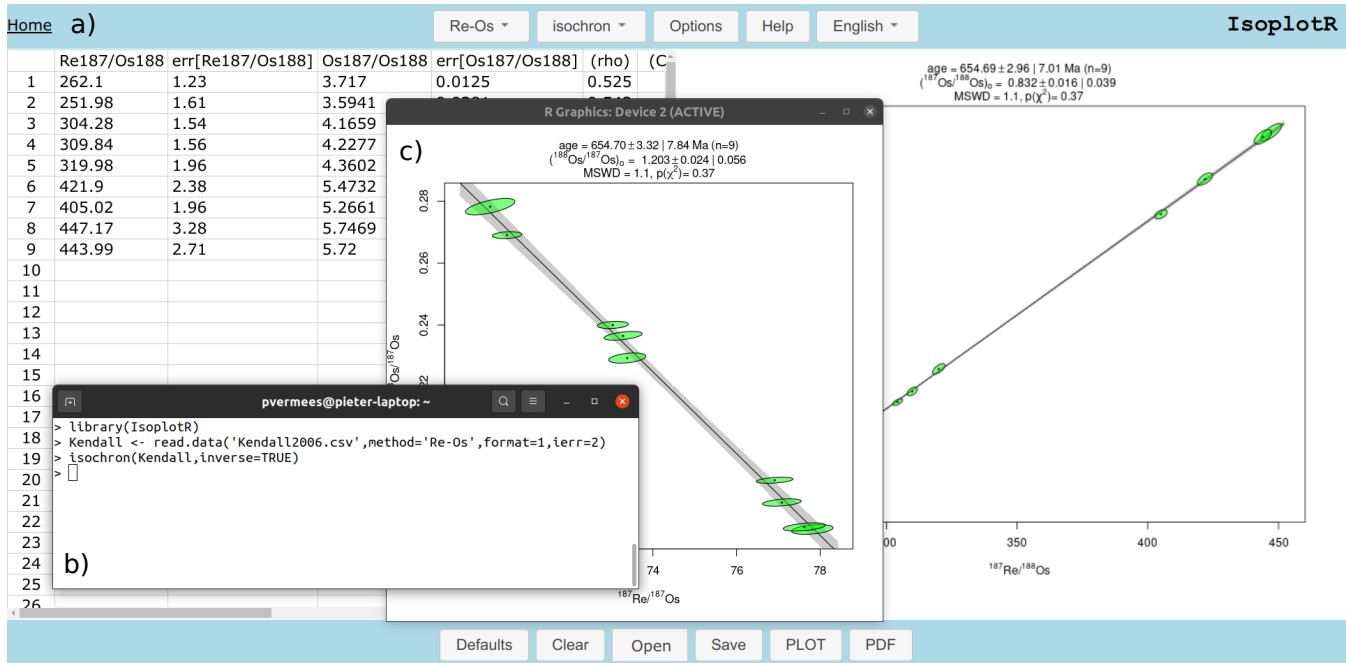

**Figure 3.** Conventional and inverse isochrons can be constructed with `IsoplotR`, (a) either using its graphical user interface, (b–c) or from the `R` command prompt. This is illustrated here for a Re–Os dataset of Kendall et al. (2006) that exhibits weaker error correlations than the example of Figure 1. The normal (a) and inverse (c) isochron produce nearly identical results.

Published datasets (which are usually tabulated in a conventional isochron format) can be re-evaluated by transforming them to inverse isochron ratios using Equation 7, either explicitly or internally within `IsoplotR`. The two isochron formulations produce identical results (Dalrymple et al., 1988) if the relative uncertainties of the ratio measurements are reasonably small ($<$ 5%, say). In the presence of larger uncertainties, inverse isochrons produce the most accurate results. We therefore recommend that inverse isochrons are used instead of conventional isochrons for Re–Os and K–Ca geochronology, and any other datasets exhibiting strong error correlations.

*Code and data availability.* `IsoplotR` is free software released under the GPL-3 license. The package and its source code are available from https://cran.r-project.org/package=IsoplotR.

*Author contributions.* PV wrote the software and the paper. YL formulated the research question and contributed to the writing of the paper.

*Competing interests.* Pieter Vermeesch is an associate editor of *Geochronology*.

*Acknowledgements.* We thank David Selby for feedback on an early version of the manuscript. This research was supported by National Key Research and Development Program of China grant #2018YFA0702600 and National Natural Science Foundation of China grant #42022022 awarded to YL; and by NERC standard grant #NE/T001518/1 ('Beyond Isoplot') awarded to PV. Donald Davis, Ryan Ickert and an anonymous reviewer are thanked for their constructive reviews, which prompted us to develop the semi-synthetic K–Ca model.

140

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
