# Peer review of "Short communication: Inverse isochron regression for Re–Os, K–Ca and other chronometers"

_Geochronology, 2021_

## Author Comment (AC1)

**Response to review by Dr. Donald Davis**

Pieter Vermeesch and Yang Li

April 7, 2021

We thank Dr. Davis for his positive and open minded review. We are happy that, after reading our manuscript, he has come to the conclusion that conventional isochrons should be avoided completely!

The reviewer is correct that the approximate symbol for the correlation coefficient in equation 4 derives from the fact that it is the first order approximation to a Taylor expansion. However, this is not the reason why the conventional and inverse isochron age estimates may disagree in the presence of large uncertainties. The actual reason is that isotopic ratios are strictly positive quantities with skewed error distributions. The weighted least squares algorithm of York (1969) does not take into account this skewness and this may cause the conventional and inverse isochron age to diverge.

To address this issue, we will replace the K–Ca example with a semi-synthetic version of it, to demonstrate that the inverse isochron is more accurate than the conventional isochron. The semi-synthetic dataset will be created as follows:

1. Let $x_i$ be the $i^{\text{th}}$ $^{40}$K/$^{44}$Ca ratio (out of $n = 30$) from the Harrison et al. (2010) dataset, and let $\sigma[x_i]$, $\sigma[y_i]$, $\rho[x_i, y_i]$ be the standard errors and error correlation of the $^{40}$K/$^{44}$Ca and $^{40}$Ca/$^{44}$Ca ratios.

2. Collect $n$ pairs of logratios $\{\ln[X_i], \ln[Y_i]\}$ from a bivariate normal distribution with means $\{\ln[x_i], \ln[y_i]\}$ and covariance matrix $\Sigma_i$ where

$$y_i = y_\circ + 0.895 x_i (\exp[\lambda_{40} t] - 1) \tag{1}$$

in which $y_\circ = 66$ is the initial $^{40}$Ca/$^{44}$Ca ratio, $t = 800$ Ma is the true K–Ca age, and

$$\Sigma_i = \begin{bmatrix} \frac{1}{x_i} & 0 \\ 0 & \frac{1}{y_i} \end{bmatrix} \begin{bmatrix} \sigma[x_i]^2 & \rho[x_i, y_i]\sigma[x_i]\sigma[y_i] \\ \rho[x_i, y_i]\sigma[x_i]\sigma[y_i] & \sigma[x_i]^2 \end{bmatrix} \begin{bmatrix} \frac{1}{x_i} & 0 \\ 0 & \frac{1}{y_i} \end{bmatrix}$$

3. The semi-synthetic dataset is then given by $\{X_i, Y_i\}$ (for $1 \leq i \leq n$) with covariance matrices $\Sigma_i'$ that are computed as follows:

$$\Sigma_i' = \begin{bmatrix} X_i & 0 \\ 0 & Y_i \end{bmatrix} \Sigma \begin{bmatrix} X_i & 0 \\ 0 & Y_i \end{bmatrix}$$

The logarithmic transformation is necessary to account for the inevitable skewness of the error distributions. Even though the semi-synthetic dataset is defined in terms of the conventional isochron equation (Eq. 1), it is the inverse isochron that most accurately estimates the age:

[Figure]

Figure 1: a) conventional and b) inverse isochron of a semi-synthetic dataset based on a K–Ca data of Harrison et al. (2010). The true age is 800 Ma and the true initial $^{40}Ca/^{44}Ca$ ratio is 66. The inverse isochron better approximates these values than the conventional isochron.

We have repeated this numerical experiment numerous times and the inverse isochron is always more accurate. A full theoretical discussion of this phenomenon falls outside the scope of our short communication. But we hope that this example satisfies the request from the reviewer, whom we would like to thank again for the useful suggestion.

**References**

Harrison, T. M., Heizler, M. T., McKeegan, K. D., and Schmitt, A. K.: In situ $^{40}K$–$^{40}Ca$ 'double-plus' SIMS dating resolves Klokken feldspar $^{40}K$–$^{40}Ar$ paradox, Earth and Planetary Science Letters, 299, 426–433, 2010.

York, D.: Least squares fitting of a straight line with correlated errors, Earth and Planetary Science Letters, 5, 320–324, 1969.

---

## Author Comment (AC2)

**Response to review by Dr. Ryan Ickert**

Pieter Vermeesch and Yang Li

April 8, 2021

The detailed review by Dr. Ickert agrees with the contents of the paper, but is critical of certain aspects of the presentation.

> *Although the authors of this manuscript appear to be aware that their work is not new (line 66) they don't make clear what differentiates this contribution from others. The manuscript would be improved if it were better able to highlight a novel contribution.*

As the reviewer notes, we did not claim to have invented a new approach to geochronology. Our short communication simply aims to draw the attention of the wider geochronology community to the benefits of inverse isochrons. The positive review by Dr. Donald Davis proves that there is a need for such a paper. The Re–Os method will benefit from a switch to inverse isochrons, and so will the K–Ca and other methods. It is true that inverse ratios are widely used in Pb–Pb, U–Pb, Ar–Ar and Th–U geochronology. In each case, geochronologists have effectively 'reinvented the wheel'. By generalising inverse isochrons to all common chronometers in `IsoplotR`, our paper will hopefully represent the last time that such reinvention is necessary. Inverse isochrons have been part of `IsoplotR` for more than a year, yet nobody seems to have noticed this feature so far. Our short paper intends to change that.

Besides this modest goal of advertising a great graphical tool, our short paper makes two further 'novel' contributions:

1. As the reviewer points out, it provides a handy formula to convert conventional isotope ratios to inverse ratios. I am not aware of this calculation being documented in the geological literature.

2. In response to the two reviews, the revised version of the paper will demonstrate that the inverse isochron produces more accurate results than the conventional isochron.

The revised abstract will highlight these two aspects of the paper.

> *The argument that it more easily allows outlier identification is not particularly compelling: The Re-Os example in Fig 1C is unconvincing – the outliers they "identify" on the plot are not clear, at least*

> *to me, and anyways that result is muddled somewhat by the fact that they have mixed samples of likely different ages on the same diagram (as described in the original paper). A better way to identify data that have undue weight on the MSWD is to simply inspect the variance normalized residuals and look for the largest values.*

The comment about the Morelli dataset being a mixture of three samples is well taken and will be acknowledged in the revised manuscript.

We attribute the dispersion to aliquots 1, 12 and 14 because their error ellipses have the least overlap with the isochron. This is how we understand most isochron users interpret their data. It may be so that inspecting the variance normalised residuals may be a better approach, but we have rarely seen this being used in practice.

> *there is a persistent belief in some workers that ages determined by one regression are better or more precise than using an inverse or vice versa (e.g., Connelly et al., 2017). This would be trivial for the authors to include, by producing regression analysis on both the isochron and its inverse and demonstrating substantive equivalence. This is complicated somewhat by the fact that the two regressions become significantly distinct with extremely large uncertainties (as they state on line 100) and also with highly overdispersed data, but it is easy to carve out that as an exception.*

We will address this request by modifying Figures 1 and 2. The new versions of these figures will report the age and intercept for both the conventional and inverse isochron, thereby highlighting the differences between them. Figure 1 will become:

[Figure]

Figure 1: a) conventional and b), c) inverse isochron of the Re–Os data of Morelli et al. (2007). The isochron ages are similar but the inherited $^{187}Os/^{188}Os$-ratios are not. Note the double x- and y-axis of panel b), which follows a suggestion from Reviewer 1.

Figure 2 will be replaced by a semi-synthetic K–Ca dataset, which will be constructed as described in the response to Reviewer 1. This example demonstrates that, in the case of imprecise dataset, the inverse isochron produces more accurate results than the conventional isochron. We will also explore the effect of overdispersion without going into too much detail.

*As written, the manuscript gives a misleading impression about the origin of correlations in isotopic and geochronological data. While*

*poor counting statistics on denominator isotopes may be important in some Ar isotope datasets, most uncertainty correlations in real, published datasets are due to other factors, such as fractionation corrections, interelement calibration, and blank corrections.*

This is definitely true for the Wetherill concordia diagram, in which the elemental fractionation between U and Pb is responsible for the error correlations. We will clarify this in the revised manuscript.

The 'Minor elements' in the second half of the review are all straightforward to address with the following exceptions:

*A spurious correlation is something more akin to the classic "pirates are causing global warming" example (and many others, cf. https://www.tylervigen.com/spurious-correlations). This word should not be used in the manuscript to describe any of the correlations, which are all real.*

In fact the word 'spurious' was meant exactly as intended by Karl Pearson (1897) in his classic paper "on a form of spurious correlation which may arise when indices are used in the measurement of organs", which is referenced in the paper. In the case of Re–Os geochronology, it is possible to observe a strong apparent correlation between the $^{187}Os/^{188}Os$ and $^{187}Re/^{188}Os$ ratio measurements when the correlation between the true atomic $^{187}Os/^{188}Os$ and $^{187}Re/^{188}Os$ ratios is in fact zero. We will add a sentence to clarify this source of apparent confusion.

*Section 4: This whole section seems superfluous. A statement at the end of the manuscript stating that "these calculations are implemented in Isoplot R" is sufficient. The paragraph and screen grab are unnecessary.*

We would like to point out that Figure 3 achieves a lot in a small amount of space:

1. It shows a second Re–Os example with weaker error correlations.

2. It shows how to perform the calculation from the command line, which may not be obvious for readers who are not familiar with `R`.

3. In the revised version of the manuscript, we will modify this figure so that it shows both the conventional and inverse isochron, which in this case produce essentially identical results due to the good precision and excellent spread of the data along the isochron.

*Line 99: What does "mathematically equivalent" mean?*

It means that, in the limit where $s[x_i]/x_i = 0$ and $s[y_i]/y_i = 0$, that the two formulations give exactly the same result. The reviewer uses the term "substantive equivalence" in his review and we would be happy to use the same term in the revised manuscript.

**References**

Morelli, R., Creaser, R. A., Seltmann, R., Stuart, F. M., Selby, D., and Graupner, T.: Age and source constraints for the giant Muruntau gold deposit, Uzbekistan, from coupled Re-Os-He isotopes in arsenopyrite, Geology, 35, 795–798, 2007.

---

## Author Response (AR1)

Prof. Pieter Vermeesch
University College London
+44 (0)20 3108 6369
http://ucl.ac.uk/~ucfbpve/

9 May 2021

Dr. Marissa Tremblay, Associate Editor, *Geochronology*

Dear Dr. Tremblay,

On behalf of Dr. Yang Li and myself, I hereby submit a revised manuscript on "Inverse isochron regression for Re–Os, K–Ca and other chronometers", which addresses all the issues raised by the three reviewers and yourself. I have outlined these in detail in the online discussion. But in short, the new version:

1. shows that the ages obtained by conventional and inverse isochron regression can significantly differ for imprecise datasets. Using a semi-synthetic K–Ca dataset, we show that inverse isochrons produce more accurate results (addresses comments by reviewers Ickert, Davis and yourself);

2. more prominently discusses inverse isochron regression in Ar–Ar and U–Pb geochronology, with additional references (Ickert and yourself);

3. clarifies that correlated uncertainties may also arise from blank corrections and calibration errors (Ickert);

4. removes all but one instance of the word 'spurious' (Ickert and yourself);

5. includes a redrafted Re–Os figure, with double x-axis labelled by $^{187}$Re/$^{187}$Os ratio as well as Re–Os age (Davis);

6. removes the basic introduction to Re–Os geochronology (Ickert);

7. stops sort of removing the three-item list of differences between conventional and inverse isochrons, as was suggested by reviewer Ickert; but does rearrange the list in order of decreasing importance according to the reviewer;

8. advocates that inverse isochrons replace conventional isochrons in future Re–Os and K–Ca studies (Davis).

I hope that you will find the revised manuscript suitable for publication in *Geochronology*.

Best wishes,

Pieter Vermeesch